# Mechanical Properties Evaluation of Three Different Materials for Implant Supported Overdenture: An In-Vitro Study

**DOI:** 10.3390/ma15196858

**Published:** 2022-10-02

**Authors:** Mona Gibreel, Leila Perea-Lowery, Lippo Lassila, Pekka K. Vallittu

**Affiliations:** 1Department of Biomaterials Science and Turku Clinical Biomaterials Centre-TCBC, Institute of Dentistry, University of Turku, 20520 Turku, Finland; 2City of Turku Welfare Division, Oral Health Care, Puolalankatu 5, 20101 Turku, Finland

**Keywords:** overdenture, implant, attachment, flexural strength, elastic modulus

## Abstract

Aim: the aim of this study was to compare the flexural strength and elastic modulus of three-dimensionally (3D) printed, conventional heat-cured, and high-impact implant-supported overdenture materials specimens. Materials and Methods: Thirty implant-supported overdenture materials specimens (bar-shaped, 65.0 × 10.2 × 5.1 ± 0.2 mm^3^) with one central hole were fabricated using 3D-printed, heat-cured conventional, and high-impact denture base resins (*n* = 10/group). Autopolymerizing acrylic resin was used to attach titanium matrix housings to the central holes of the specimens. A three-point bending test was conducted using a universal testing machine and a model analog with a crosshead speed of 5 mm/min. The indicative flexural strength and elastic modulus were recorded. Data were statistically analyzed using analysis of variance (ANOVA) and the Tukey tests at α = 0.05. Results: One-way ANOVA revealed a significant effect of denture base material on the flexural strength (*p* < 0.001) but not on the elastic modulus (*p* = 0.451) of the evaluated materials. The flexural strength of the 3D-printed specimens (95.99 ± 9.87 MPa) was significantly higher than the conventional (77.18 ± 9.69 MPa; *p* < 0.001) and high-impact ones (82.74 ± 7.73 MPa; *p* = 0.002). Conclusions: The maximum flexural strength was observed in the 3D-printed implant-supported overdenture material specimens, which might indicate their suitability as an alternative to the conventionally fabricated ones. Flexural strength and elastic modulus of conventional and high-impact heat-cured implant-supported overdenture materials specimens were comparable.

## 1. Introduction

Poly(methyl methacrylate) (PMMA) has been widely used as a denture base material due to its ease of manipulation, pleasing cosmetic-esthetic outcomes, and affordable cost [1,2]. Denture base fractures due to fatigue of the material and impact forces are common, and some improvement to the material has taken place [3,4,5,6]. To enhance its mechanical properties, PMMA polymer has been modified by adding rubber compounds or acrylic-elastomer copolymers to the powder to produce high-impact acrylic resins [7,8]. These rubber compounds can absorb the crack energy and arrest or slow its propagation through the denture base [9,10,11].

PMMA denture bases can be fabricated conventionally by compression molding, which involves a series of multiple clinical and laboratory steps [12]. Recently, the processing of dental materials with computer-aided design and computer-aided manufacturing (CAD-CAM) based techniques, such as subtractive milling (SM) and additive manufacturing (AM), has enhanced their precision while reducing the time and technical sensitivity of the laboratory procedures [13,14,15]. In 1994, Maeda et al. [16] presented the first CAD-CAM complete removable dental prosthesis. Thanks to this technology, it is possible to fabricate complete dentures in just two clinical visits [17].

Implants can be utilized in completely edentulous patients to improve denture retention, masticatory efficiency, maximum bite force, and patient satisfaction while minimizing residual ridge resorption [18]. Solitary and bar-type attachments are available for dental implant overdentures [19]. Solitary attachments, such as balls, magnets, locators, and OT Equators, are easier to maintain and can be used in limited inter-arch spaces, while bars provide more stability to overdentures [20,21]. Usually, each attachment system has two components, one is embedded within the denture base while the other is screwed to the implant [22]. In long-term evaluations, implant overdentures demonstrated high implant and prosthetic survival rates, few complications, high patient satisfaction, and favorable biological parameters [23]. However, on the other hand, considering the higher occlusal force exerted by the implant support, which may surpass the yield stress of the material, overdenture fracture during functioning is prevalent in clinical practice [24,25,26,27]. The majority of fractures were noticed in the thinner denture base area surrounding the abutment [26,27]. While the flexural strength of 3D-printed denture base materials has been reported [28,29,30], information concerning the flexural strength of 3D-printed implant-supported overdentures is, to the authors’ knowledge, not available in the literature.

Therefore, the aim of this study was to compare the flexural strength and elastic modulus of 3D-printed, conventional heat-cured, and high-impact implant-supported overdenture materials. The hypothesis was that implant-supported overdentures processed from different materials would display similar flexural strength and elastic modulus.

## 2. Materials and Methods

Three different commercially available denture base materials were evaluated in this study: a 3D-printing resin, a high-impact heat-cured acrylic resin, and a conventional heat-cured acrylic resin (Table 1).

Thirty specimens (65.0 × 10.2 × 5.1 ± 0.2 mm^3^; *n* = 10/material) mimicking overdentures with a central hole (10.0 × 6.0 × 3.5 mm^3^) were fabricated. The 3D-printed denture base specimens were designed using a CAD program software AutoCAD (Autodesk Inc.; San Francisco, CA, USA) and exported as a standard tessellation language (STL) file. A digital light processing (DLP) 3D printer (Asiga MAX™; Asiga, Sydney, Australia) was used to print the specimens with a layer thickness of 50 μm in a vertical orientation on the printing platform and identical supports were generated for all the specimens. The post-processing of the printed specimens was done according to the manufacturer’s guidelines by cleaning them in isopropanol (>98%) for two minutes inside an ultrasonic path (Form Wash; Formlabs, Berlin, Germany) to dissolve any unpolymerized resin and then rinsing in a clean alcohol solution for an additional three minutes. The specimens were allowed to dry in the air and then post-cured in a polymerization unit at 60 °C for 30 min (Form cure; Formlabs, Berlin, Germany). All supports were then removed. In order to fabricate the conventional and high-impact resin specimens, custom Teflon molds were made by using additional 3D-printed specimens as a template. The materials were mixed, packed into the molds, and finally polymerized according to the manufacturer’s instructions as described in Table 1. Successive polishing of the specimens was carried out with a rotary polishing device (LabPol-21; Struers, Ballerup, Denmark) using 800- and 1200-grit FEPA abrasive papers (Buehler, New York, NY, USA). In this study, Novaloc titanium matrix housings (Valoc, Möhlin, Switzerland) (5.5 mm in diameter and 2.3 mm in height) with white (light) retention inserts were used. Specimens’ holes were first coated with a monomer liquid, then filled with an autopolymerizing PMMA resin mixture (Palapress; Kulzer GmbH, Hanau, Germany) before being slipped over the matrix housings to insert them into the specimen. After that, specimens were processed in distilled water at 55 °C and under air pressure of 300 kPa for 15 min in a pneumatic polymerizing unit (Ivomat IP3; Ivoclar Vivadent AG, Schaan, Liechtenstein) to polymerize the pickup material (Figure 1). The specimens were stored in dark containers filled with distilled water at 37 °C for 30 days before testing.

Indicative flexural strength (FS) and elastic modulus (EM) of the specimens were determined by a 3-point bending test carried out using a universal testing machine (Lloyd model LRX; Lloyd Instruments, Bognor Regis, UK). The distance between the test supports was set at 50 mm, while the load was applied at a crosshead speed of 5 mm/min using an analog [31] (Novaloc Model Analog; Valoc, Möhlin, Switzerland) for load application (Figure 2). The load-deflection curves were recorded using analysis software (Nexygen 4.0; Lloyd Instruments). FS and EM were automatically calculated by the machine software using Formulas (1) and (2) [30]:FS (MPa) = 3FL/2bh^2^(1)
EM (MPa) = F_1_L^3^/4bh^3^d(2)
where F is the maximum load (N), L is the span length (mm), b is the specimen width (mm), h is the specimen thickness (mm), F_1_ is the load (N) at a point on in the straight-line segment of the load-deflection curve, and d is the recorded deflection (mm) at load F_1_.

Statistical software (SPSS V25; IBM Corp., Armonk, NY, USA) was used to analyze the data using one-way ANOVA. If statistically significant, resolution of the significance factor was achieved by pairwise comparisons with Tukey’s post hoc analysis (α = 0.05).

## 3. Results

All the manufactured specimens underwent the test and displayed a complete fracture. Figure 3 shows the mean values of flexural strength and elastic modulus. The one-way ANOVA revealed a significant effect of the denture base material on the fractural strength (*p* < 0.001) but not on the elastic modulus (*p* = 0.451) (Table 2).

The flexural strength of the FREEPRINT denture (95.99 ± 9.87 MPa) was significantly higher than Paladon 65 (77.18 ± 9.69 MPa; *p* < 0.001) and Lucitone 199 (82.74 ± 7.73 MPa; *p* = 0.002). The flexural strength difference between Paladon 65 and Lucitone 199 denture base materials was not significant (*p* = 0.164). Figure 4 depicts the load-deflection curves of the tested materials, which highlighted the wider plastic deformation of the high-impact and the 3D-printed resins when compared to the conventional heat-cured PMMA. Paladon 65 and Lucitone 199 specimens displayed fracture in the middle above the attachment matrix, while FREEPRINT denture specimens displayed fracture in multiple sites in addition to the middle one (Figure 5).

## 4. Discussion

The study hypothesis was partially rejected, as implant-supported overdenture specimens processed from different materials displayed significant differences in terms of their flexural strength, while the modulus of elasticity was not significant.

Denture base fractures account for approximately 9.3% to 21.4% of complications in implant-supported overdentures [32]. Fractures are more frequent in the area around the implants or abutments due to stress accumulation and denture base deformation in this thinner area [24,33]. Flexural strength is a key property that provides information about the ability of a denture base material to withstand functional masticatory loads, as it replicates the type of force applied to the denture during mastication [19,34].

Test specimens were stored in water for 30 days before testing to simulate the wet oral environment. Water storage is a time-dependent process that results in the leaching of soluble components such as unreacted polymers and plasticizers from denture base resins, causing micro-voids formation and inward water diffusion, which negatively affect the polymer strength [35].

The results of the study revealed that 3D-printed implant-supported overdentures had a significantly higher flexural strength than those made from conventional and high-impact heat-cured acrylic resin, while the increase in elastic modulus was not significant. Additive manufacturing is an automated process with a low human error rate and a standardized workflow that employs resins with different compositions [29]. Materials for DLP are usually based on a variety of methacrylate or acrylate monomers and oligomers. However, usually manufacturers keep specific components confidential since they belong to their proprietary information. While the flexural strength of 3D-printed implant-supported overdentures was not available in the literature, studies comparing the flexural strength between conventional and 3D-printed denture base materials revealed some variations in results. Some studies [15,27,28,36,37] concluded that 3D-printed denture base resins were inferior to conventional heat-cured ones in terms of their mechanical properties, while they were equivalent in other studies [29]. Interestingly, a 3D-printed denture base material from VOCO GmbH (V-Print dentbase) recorded a flexural strength of up to 171 MPa in a recent study by Li et al. [29], suggesting a continuous improvement in denture base resins used for additive manufacturing. Differences in material composition [38,39] and degree of double-bond conversion [40] across different resin brands have implications on the mechanical characteristics of resins used for 3D printing. Other variables such as printers, printing orientation, and post-curing techniques would also affect the mechanical properties of the 3D-printed objects [38,41,42,43].

The 3D-printed overdenture specimens investigated in this study showed greater plastic elongation followed by the high-impact, while the conventional heat-cured resin specimens exhibited only elastic elongation (Figure 4). The difference in elongation behavior could be related to the different manufacturing techniques used, whether the resin is deposited into a mold and entirely polymerized in one step or whether it is subjected to multiple polymerizations during the deposition of several resin layers [29]. While in high-impact specimens, it could be due to the material’s rubber content, which absorbs more energy and makes the material less brittle [11].

The type of material used to secure the attachment housing, as well as the bonding between the denture base and the pickup material, can affect the flexural strength of the overdenture [25,44]. In the current study, autopolymerizing resin was employed as the pickup material. The attachment space within the denture base was first etched with methyl methacrylate (MMA) monomer. Chemical etching with the MMA monomer would substantially improve the bond strength between the denture base and the repair resin by dissolving conventional denture base materials, allowing the monomer to penetrate the surface and create an interwoven network [2,45]. In addition, a previous study [13] investigating the repairability of a 3D-printed denture base material, FREEPRINT denture, found that a cohesive failure mode prevailed in unaged specimens etched with the liquid monomer of Palapress. This finding was explained by the ability of MMA to dissolve the surface of the 3D-printed denture base material, penetrate the polymer network, and crosslink with the unsaturated carbon-carbon double bonds of the denture base polymer [13].

The specimens used in this study deviated from standard denture configurations, and only one 3D-printed denture base material was investigated. Within these limitations, the results showed that the 3D-printing denture base material, FREEPRINT denture, outperformed the conventional and high-impact heat-cured PMMA resins, Paladon 65 and Lucitone 199, in terms of their flexural strength when used for implant-supported overdentures. Accordingly, making use of FREEPRINT denture might minimize the incidence of overdenture fracture. In addition, since the material is MMA-free, according to the manufacturer’s claims, it offers an alternative to PMMA denture base resins for patients with sensitivity or allergy to MMA. Finally, 3D-printed implant-supported overdenture materials can be considered as an appropriate candidate to substitute the heat-cured resins in terms of flexural strength and elastic modulus. However, more studies are needed to consider 3D-printed denture base materials as a viable alternative in all aspects of implant-supported overdenture materials. Also, the use of high-impact resin did not lead to a significant increase in the flexural strength of implant-supported overdentures, making it unnecessary given that it increases the cost of the prosthesis.

## 5. Conclusions

Based on the results of this in vitro investigation, the following can be concluded:3D-printed implant-supported overdenture material specimens displayed the highest flexural strength while being significantly different from those made of conventional and high-impact heat-cured acrylic resin.Conventional and high-impact heat-cured acrylic resin denture base materials displayed comparable flexural strength and elastic modulus when used for the fabrication of implant-supported overdenture specimens.

## Figures and Tables

**Figure 1 materials-15-06858-f001:**
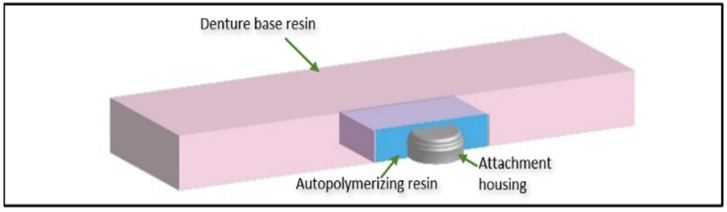
Schematic of test specimen longitudinal cross section with pickup material and attachment housing.

**Figure 2 materials-15-06858-f002:**
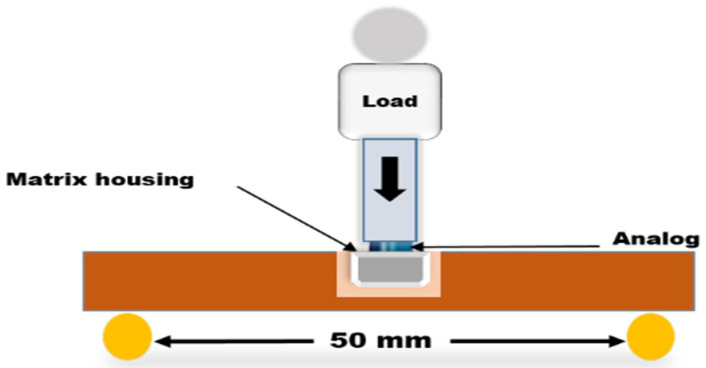
Schematic diagram of three-point bending test procedures.

**Figure 3 materials-15-06858-f003:**
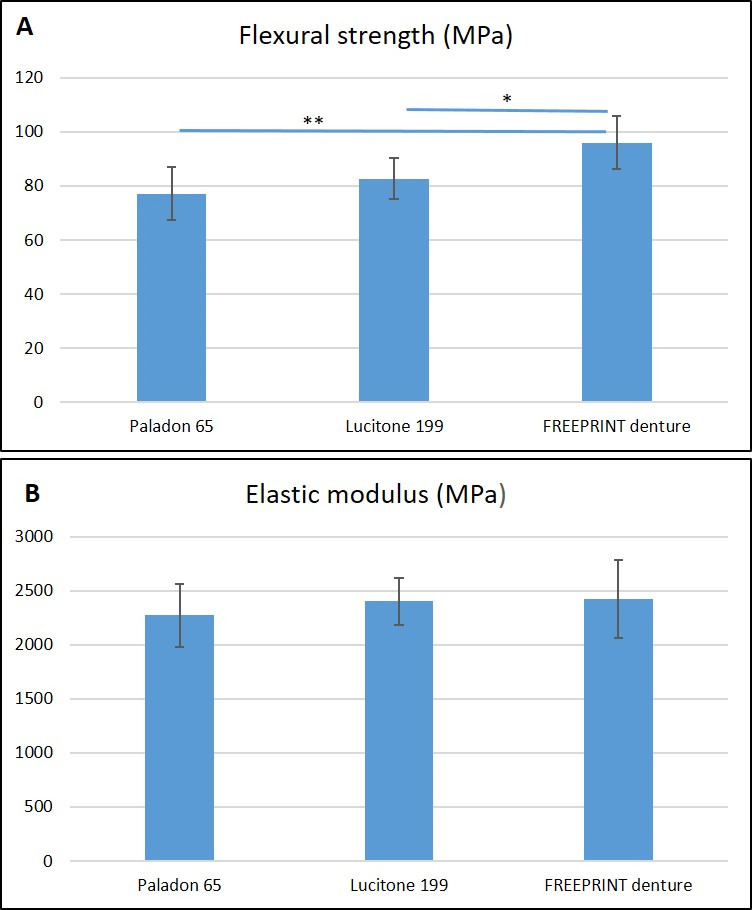
Mean values of (**A**), Flexural strength and (**B**), Elastic modulus of evaluated overdenture base materials. * and ** indicate a statistically significant difference by Tukey HSD (*p* < 0.05). Error bars represent standard deviation (SD).

**Figure 4 materials-15-06858-f004:**
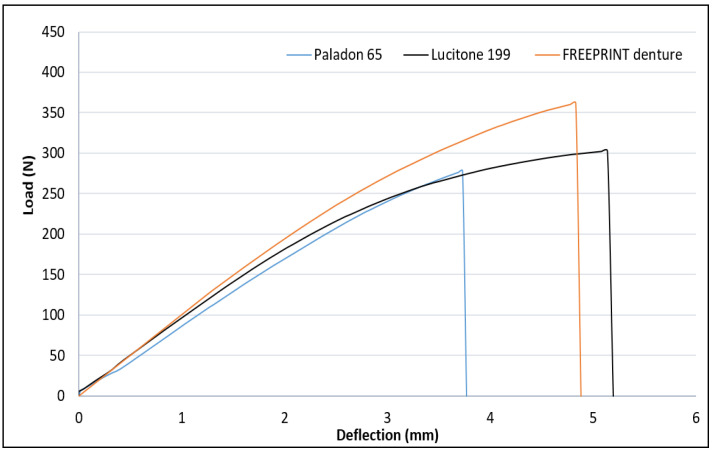
Load deflection curves of evaluated overdenture base materials.

**Figure 5 materials-15-06858-f005:**
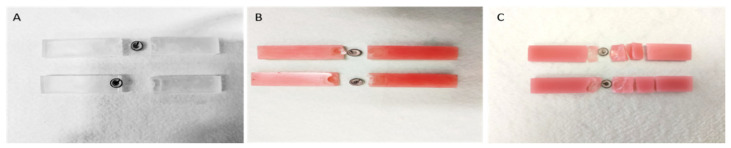
Fractured specimens of evaluated overdenture base materials. (**A**), Paladon 65; (**B**), Lucitone 199; (**C**), FREEPRINT denture.

**Table 1 materials-15-06858-t001:** Materials used in study identified by material commercial name, description, manufacturer, processing method, and chemical composition.

Material	Description	Manufacturer	Processing Method	Chemical Composition.
Paladon 65	Heat-cured acrylic resin	Kulzer GmbH, Hanau, Germany	20 min at 90 °C, then cool down slowly in the water bath.	Liquid: methylmethacrylate (>90%); tetramethylene dimethacrylate (≥1–≤5%); p-Mentha-1,4-diene (<0.25%) Powder: (based on methacrylate copolymonomers) methylmethacrylate (≥1–≤5%); dibenzoyl peroxide (≥0.25–<1%)
Lucitone 199	Heat-cured high-impact acrylic resin	Dentsply Intl, York, Pa	90 min at 70 °C and 30 min in boiling water	Powder: PMMA (with rubber molecules) 95–100% Liquid: methyl methacrylate (80–100%), ethylene dimethacrylate (1–20%)
FREEPRINT denture	Light curing resin for 3D-printing	Detax, Ettlingen, Germany	3D printing in a DLP printer 385 nm, post-curing in a light chamber for 30 min at 60 °C	MMA-free Mixture of acrylic/methacrylic resins with auxiliary matters
Palapress	Autopolymerizing resin (pick up material)	Kulzer GmbH, Hanau, Germany	15 min at 55 °C and 300 KPa	Liquid: methylmethacrylate (>90%); tetramethylene dimethacrylate (≥1–≤5%); maleic acid (<0.1%); 2-Hydroxy-4-methoxy benzophenone (<0.25%); mequinol (<1%); Quaternary ammonium compounds, tri-C8-10-alkylmethyl, chlorides (≥0.025–<0.25%) Powder: (based on methacrylate copolymers) dibenzoyl peroxide (≥1–<2.5%); methyl methacrylate (≥1–≤5%); 1-Benzyl-5-phenylbarbitursäure (≥0–≤5%)

**Table 2 materials-15-06858-t002:** ANOVA statistics for flexural strength and elastic modulus.

ANOVA						
		Sum of Squares	df (Degree of Freedom)	Mean Square	F	*p* Value
Flexural strength	Between groups	2056.427	2	1028.214	12,280	<0.001
	Within groups	2511.834	30	83.728		
	Total	4568.261	32			
Elastic modulus	Between groups	142,880.893	2	71,440.446	0.819	0.451
	Within groups	2,618,450.946	30	87,281.698		
	Total	2,761,331.839	32			

## Data Availability

The data presented in this study are available on reasonable request from the corresponding author.

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
