# Peer review of "Mechanical Properties Evaluation of Three Different Materials for Implant Supported Overdenture: An In-Vitro Study"

_materials, 2022, doi:10.3390/ma15196858_

Round 1

Reviewer 1 Report

In this manuscript, the authors evaluated the flexural and elastic modulus of three different implant-supported overdenture materials: 3D printed, conventional heat-cured and high impact heat-cured acrylic resin. Data obtained were statistically analysed and described. Furthermore, the submitted manuscript is within the scope of the Materials journal. Thus, I suggested it to be accepted as short communications after some revisions.

Comments

-          The tittle of the manuscript is quite generic. Authors compare different type of resins, not just 3D-printed ones.

-          The section Materials and Methods should be better organized. First, description and characteristics of materials employed need to be described. Then, in another paragraph, explain sample preparation for the analysis (including at the end the Figure 1). Finally, describe the methods employed for sample characterizations, i.e., the flexural strength (FS) and the elastic modulus (EM).

-          Why samples were processed in distilled water at 55ºC and not other temperature? What means processed at 300 kPa for 15 min? There is a pression that needs to be applied? Please explain better in Materials and Methods section.

-          Why samples are stored at 37ºC? And why samples are stored in distilled water? I understand that water stimulate the wet oral environment but other solution more similar to the human saliva should be better as other factors such as pH, presence of some salts, etc. can change the nature of the cured materials and, therefore, the mechanical properties. Furthermore, no indications about light environment during sample storing are written. Light, even in samples already postcured, can have influence on the physico-chemical properties.

-          There are few results to confirm and conclude that “3D-printed implant-supported overdenture material specimens can be used as a viable alternative to those made of heat-cured resins”. In terms of flexural strength and elastic modulus, they can be considered appropriate candidates to substitute the heat-cured resins. However, more studies are needed to consider 3D-printed materials as a viable alternative in all aspects of the implant-supported overdenture materials.

-          Despite the manuscript is correctly written, it presents some typos and writing error. A proofread would be required:

o   Table foot of Table 1: *DLP: digital light processing. Why this is needed? Is just written along the text. Further, there is not * symbol in the table neither in the text.

o   Line 85 to 86: “…(Figure 1) Holes were first…”. A dot is missing after the parentheses.

o   Line 92: “The specimens were stored in distilled water at 37°C for 30 days before testing.”. This sentence should be included in the previous paragraph due to is related with sample preparation and not sample characterization.

o   Line 60: Materials and Methods sections is missing the number: 2. Materials and Methods.

o   Lines 191 to 193. Different font was used. Change and put the same along all the manuscript.

o   Quality of figures should be improved. Further, Figure 2 should include part a) and b) in order to differentiate the two plots included. Why FREEPRINT denture flexural strength is displayed in green?

Reviewer 2 Report

Dear authors, thanks to provide this research.

In general, I think this study is merit of "Article" instead of "Short communication". I will suggest to the editor. You propose several tests and an hypothesis. Nevertheless, in order to improve the quality of the manuscript, some modifications are needed.

English language needs several improvements. I suggest a professional editing.

For example, the title is: "3D-printed implant-supported overdenture materials mechanical properties. A short communication".

 but it should be: "Mechanical properties evaluation of there different materials for implant-supported overdenture: an in-vitro study"

Introduction

The introduction section is well presented. In the third paragraph, lines 46 to 47 I suggest to add and discuss the following papers to improve the clinical importance of this research:

Tallarico, M.; Cervino, G.; Montanari, M.; Scrascia, R.; Ferrari, E.; Casucci, A.; Xhanari, E.; Lupi, S.M.; Meloni, S.; Ceruso, F.M.; Rodriguez y Baena, R.; Cicciù, M. OT-Equator® Attachments Comparison for Retaining an Early Loaded Implant Overdenture on Two or Three Implants: 1 Year RCT Preliminary Data. Appl. Sci. 2021, 11, 8601. https://doi.org/10.3390/app11188601

Montanari, M.; Scrascia, R.; Cervino, G.; Pasi, M.; Ferrari, E.; Xhanari, E.; Koshovari, A.; Tallarico, M. A One-Year, Multicenter, Retrospective Evaluation of Narrow and Low-Profile Abutments Used to Rehabilitate Complete Edentulous Lower Arches: The OT Bridge Concept. Prosthesis 2020, 2, 352-361. https://doi.org/10.3390/prosthesis2040033

Tallarico, M.; Ortensi, L.; Martinolli, M.; Casucci, A.; Ferrari, E.; Malaguti, G.; Montanari, M.; Scrascia, R.; Vaccaro, G.; Venezia, P.; Xhanari, E.; Rodriguez y Baena, R. Multicenter Retrospective Analysis of Implant Overdentures Delivered with Different Design and Attachment Systems: Results Between One and 17 Years of Follow-Up. Dent. J. 2018, 6, 71. https://doi.org/10.3390/dj6040071

Materials and methods

In this section, you should add information (fabrication methods) for all the three specimens, and the section should be structured (location where the specimens has been fabricated, specimens' characteristics, description of the test, outcomes, statistical informations… How analyzed the specimens and make the test…) Please follows the CRIS guidelines in order to improve the quality of the manuscript.

https://www.ncbi.nlm.nih.gov/pmc/articles/PMC4127685/

Results

Results section should be improved adding some informations about the specimens. Did all the specimens underwent the tests? There were some deviation from the original protocol? Please check the CRIS guidelines.

Moreover, some pictures of the samples could be added to make more clear the study for the readers.

Discussion

Lines 191-193 atre in a different font size.

In the discussion section the authors can add some clinical implications of this research, such us the point number 2 of the conclusions.

Conclusions 

The point number 2 must to be moved in the discussion section. In the conclusions only the proved statements must to be reported.

Reviewer 3 Report

In this research article Thirty implant-supported overdenture 10 materials with one central hole were fabricated using 3D-printed, heat-cured conventional, and high-impact denture base resins were studied. This study compared Flexural strength and elastic modulus of conventional and high impact heat-cured implant-supported overdenture materials specimens. The following are some of the major and minor correction need to be updated in the revised manuscript.

Major Corrections:

1.       Can author add the 3-point bending testing experiment is performed using which standards.

2.       In ANNOVA study need to add the table which shows sq mean, R sq. and p value for the variable consider under study.

Minor Corrections:

1.       From Figure 1. Schematic of test specimen cross section with pickup material and attachment housing, it is not clear along which plane or axis the cross section is taken.  Define the coordinate system and specify the cross-section plane.

2.       Figure 1,2 and 3 shows variation of different font type used, need to replot with same font type as per journal template.

3.       Experimental setup image needs to add in a specific section of the research article.

4.       Figure 2. Mean values of flexural strength and elastic modulus of evaluated overdenture base materials shows in two different unit.  Must be specified in one of the standard units. (MPa or GPa)

5.       Line no.131 spacing between value and symbols required.

6.       Under ANNOVA, which type of cure fitting is used in regression analysis.

7.       Line no. 191-193 statement should be written on the Journal standard format.  No need to change the font size.

The manuscript can be accepted after major and minor corrections in the revised manuscript.
